# Prospects for a prolonged slowdown in global warming in the early 21st century

Thomas R. Knutson[1], Rong Zhang[1] & Larry W. Horowitz[1]

Global mean temperature over 1998 to 2015 increased at a slower rate ($0.1\,K$ decade$^{-1}$) compared with the ensemble mean (forced) warming rate projected by Coupled Model Intercomparison Project 5 (CMIP5) models ($0.2\,K$ decade$^{-1}$). Here we investigate the prospects for this slower rate to persist for a decade or more. The slower rate could persist if the transient climate response is overestimated by CMIP5 models by a factor of two, as suggested by recent low-end estimates. Alternatively, using CMIP5 models' warming rate, the slower rate could still persist due to strong multidecadal internal variability cooling. Combining the CMIP5 ensemble warming rate with internal variability episodes from a single climate model—having the strongest multidecadal variability among CMIP5 models—we estimate that the warming slowdown ($<0.1\,K$ decade$^{-1}$ trend beginning in 1998) could persist, due to internal variability cooling, through 2020, 2025 or 2030 with probabilities 16%, 11% and 6%, respectively.

[1] Geophysical Fluid Dynamics Laboratory/NOAA, 201 Forrestal Road, Princeton, New Jersey 08540, USA. Correspondence and requests for materials should be addressed to T.R.K. (email: Tom.Knutson@noaa.gov).

Global mean surface temperature has increased since the mid 1800s (ref. 1), mainly during two multidecadal periods: early 20th century (1908 to 1941) and late 20th century (1973 to 2001). Between the two warming epochs was a mid-20th century warming hiatus lasting over 30 years (about 1941 to 1973). Since 2000, temperatures have not warmed as rapidly as during the late 20th century[1–3]. This latter warming slowdown is minor compared with the century-scale warming, and is much shorter in duration than the 1941–1973 hiatus[3]. Nonetheless, it is of considerable interest, and its interpretation has been debated[2,3].

Ensemble-mean global temperature anomalies from the Coupled Model Intercomparison Project 5 (CMIP5) models' historical climate forcing experiments[4] are in very good agreement with the observed late 20th century warming, and also good agreement with the 1941–1973 hiatus[3]. Several generations of climate models, as cited in the last three Intergovernmental Panel on Climate Change Assessment Reports[1,5,6] have produced a hiatus around 1940–1970 in response to a combination of anthropogenic (greenhouse gases, aerosols, etc.) and natural (volcanic eruptions, solar variability) forcing agents. Those experiments suggest that the observed 1941–1973 hiatus and late 20th century warming result predominantly from external forcing. Observed early 20th century warming[1] was more rapid than in the CMIP3 or CMIP5 models[1,6], supporting earlier findings[7] and suggesting a role for internal (unforced) climate variability in that global warming feature[8].

The post-2000 global warming slowdown has been assessed as statistically inconsistent with the continued warming in CMIP5 models by some studies[3,9,10], while others studies suggest the apparent discrepancy results from factors such as misrepresentations of forcing[11,12] or the phase of internal variability[13–15] in the real world versus CMIP5 models. Various factors that could lead to model-observation inconsistencies include: missing volcanic eruptions[16], stratospheric water vapour changes[17], missing data in warming regions like the Arctic[18], and other data adjustments[2]. Regional trend patterns (1990s to 2013) show a strong Pacific cooling[19] resembling the negative phase of the Interdecadal Pacific Oscillation[20] with anomalously strong trade winds possibly linked to earlier warm anomalies in the Atlantic[21,22]. Aspects of 20th century global temperature evolution have been simulated[23] using a climate model with eastern equatorial Pacific SSTs constrained towards observations. The Atlantic basin has also been proposed[24] as an important region for heat uptake during the recent global warming slowdown. Previous work has shown[25] that by manipulating a climate model's surface heat budget in the Atlantic basin only, northern hemisphere multidecadal variability similar to detrended observations could be reproduced. Some individual simulations from CMIP5 models can reproduce a slowdown similar to that observed[14] and some aspects of the slowdown might have been predictable[15]. A semi-empirical method[26] suggests that the contribution of northern hemispheric internal multidecadal variability to global temperature likely peaked before 2000 and has been offsetting anthropogenic warming since then.

Here we explore how long an early 21st century hiatus or slowdown could last, assuming a strong internally generated cooling superimposed upon ongoing anthropogenic warming. In a previous analysis[27], the likelihood of a hiatus (zero or negative trend in global temperature) lasting 10 or 20 years in the presence of an underlying forced warming trend of 0.2 K per decade was estimated as ~10% or <1%, respectively, with the conditional probability of an existing 15 year hiatus lasting 20 years of up to 25%. We focus here on the period 1998–2015—during which the observed trend was about 0.1 K decade$^{-1}$—as the conditional target for simulations to match. Thus we are able to include the influence of the recent record warm year (2015) on the conditional probabilities and explore the influence of possible lower transient climate response (TCR) on future trends. We also explore possible internal variability contributions to regional temperature trends in other 20th century accelerated warming and hiatus epochs. We focus on internal variability simulated by the Geophysical Fluid Dynamics Laboratory (GFDL) CM3 model control run, which simulates the strongest multidecadal internal variability among the CMIP5 models we examined. We find that there are some conditions under which a continued global warming slowdown could occur over the next decade or more, due to either an overestimate of the TCR by CMIP5 models, or due to a partial offsetting of a relatively strong forced warming signal (as simulated by CMIP5 models) by an internal variability cooling episode lasting up to a few decades.

## Results
**Global temperature internal variability estimates.** The major features of observed global mean temperature evolution since the late 1800s, discussed briefly in the introduction, are summarized in Fig. 1, along with a comparison of observed global temperature to the CMIP5 multi-model ensemble mean global temperature simulation (Methods). For discussion purposes, in this study, we divide global temperature evolution into the five general epochs, which are labelled in Fig. 1. Comparison of CM3's control run with observations (Fig. 2) confirms that the observed 20th century global warming was highly unusual compared with CM3's internal variability: there are no periods of century-scale global warming or cooling in a 5000-year sample from the CM3 control run that are comparable to the observed 20th century warming. An observed internal variability estimate for global mean temperature can be obtained by subtracting the CMIP5 All-Forcing ensemble mean from observations (Fig. 2a, blue line). This observed estimate has a smaller characteristic magnitude than much of the typical variability within the CM3 control simulation. However, the internal multidecadal variability in the CM3 model varies considerably during the 5200-year control

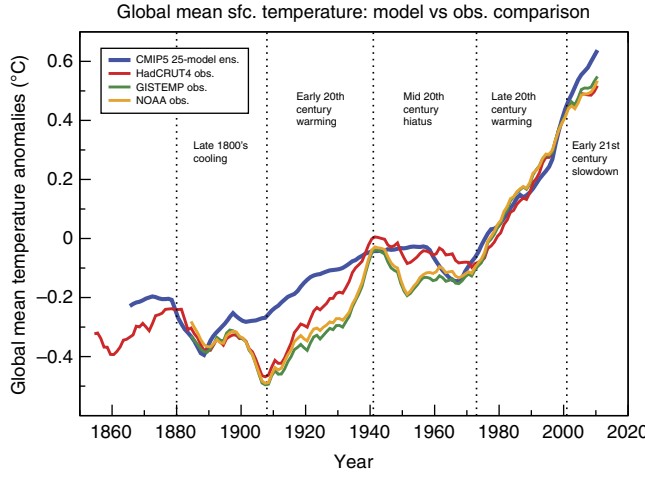

**Figure 1 | Global mean temperature anomalies in observations compared with models.** Ten-year running mean anomalies (°C) relative to a 1961–1990 reference period for observed data sets HadCRUT4.4 (ref. 50) (red line); GISTEMP[51] (green line), U.S. National Oceanographic and Atmospheric Administration (NOAA[2], orange line) or the CMIP5 multi-model ensemble (25 models) of All-Forcing historical simulations[4] (blue line). Five epochs discussed in the paper are labelled, separated by dotted vertical lines.

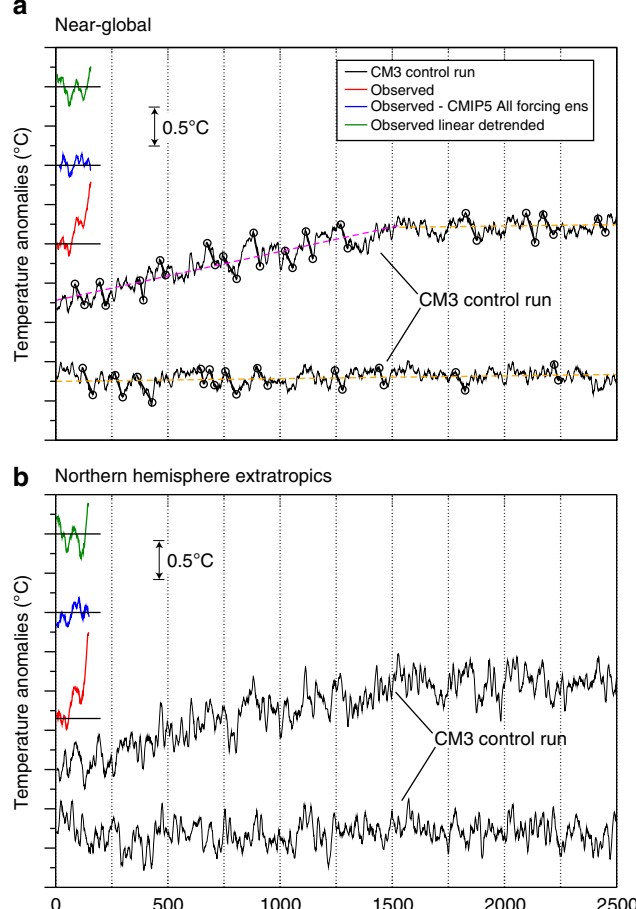

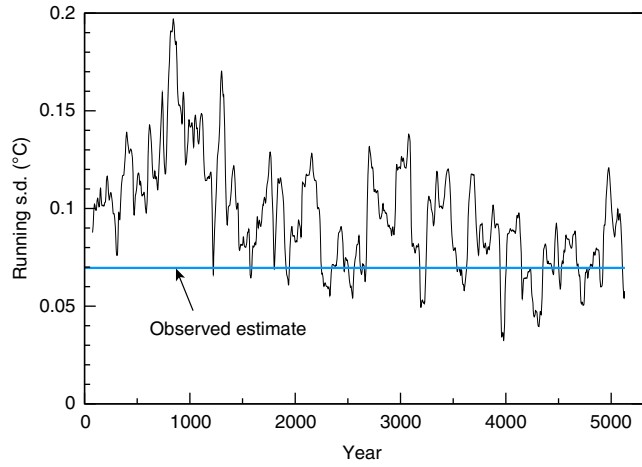

**Figure 3 | Running standard deviation of global temperatures from the CM3 control run compared with observations.** The running standard deviations from CM3 are based on overlapping 141-year segments of 10-yr running mean global surface air temperatures. A long-term drift in the control run temperatures is filtered out by removing separate linear trends over years 1–1530 and 1531–5200 and recombining the residuals to form a single 5200-year series. The drift is removed, followed by computation of 10-year running means, followed by computation of the running standard deviations of the overlapping 141-year smoothed time series segments. The blue line depicts the s.d. of 10-year running mean residual anomalies for observations (that is, observed minus CMIP5 All-Forcing ensemble mean, 141-year smoothed series). The figure illustrates the time-varying nature of the interdecadal variability in the CM3 control run. Unit: °C.

**Figure 2 | Surface temperature anomalies from a 5200-year CM3 control run compared with observations.** Anomalies (°C) averaged over (**a**) near global domain, 60°S-80°N or (**b**) northern hemisphere extratropics, 23.6°N-90°N. Ten-year running means are shown, with an arbitrary vertical shift for display purposes (see 0.5 °C vertical scale; tic marks on vertical axis at 0.25 °C intervals). CM3 control run 2 m air temperature data (black lines) are averaged over the full domain noted, without screening out data where observations are missing. The red, blue and green lines on each plot depict HadCRUT4 observed combined SST/surface air temperature anomalies over available data regions for full observations (red), observed minus CMIP5 All-Forcing ensemble mean (blue) and observed minus linear trend (green). Open circles connected by straight line segments in **a** show the 25 CM3 control run multidecadal cooling events selected for further analysis in our study. The purple and orange dashed lines in **a** depict the two linear trends fit to separate sub-segments of the 5200 year CM3 control run (purple for years 1–1530 and orange for years 1,531–5,200), which were subtracted from the control run data to approximately filter out a spurious long-term drift signal. The upper black lines in **a**,**b** depict years 1–2500, and the lower black lines depict years 2501–5000 of the CM3 control run.

integration: the running s.d. of different 141-year segments of low-pass filtered data from the CM3 control run is compared with the s.d. of a single observed 141-year segment in Fig. 3. During some multi-century periods in the CM3 run, the internal variability is at least twice as large as the observed estimate, and in a few periods it is even smaller than the observed residuals. The wide range of multidecadal variability within the 5200-year control run cautions against relying solely on ∼160 years of observations to evaluate modelled global temperature

variability. The observed internal variability estimate also has uncertainties due to observational limitations and imperfect knowledge of historical forcings and the climate response to these forcings. An alternate observed internal variability estimate, calculated as a residual from a simple linear trend, is shown for comparison in Fig. 2. The northern hemisphere extratropics (Fig. 2b) in CM3 show larger multidecadal fluctuations than the near-global temperatures.

The CM3 model has stronger global mean internal multi-decadal climate variability than the other CMIP5 models we examined (Supplementary Figs 1 and 2), and also a higher than average TCR compared with other CMIP5 models (Supplementary Fig. 2). However, we find only a weak statistical relationship across models (correlation of 0.23) between internal multidecadal variability and TCR across 21 CMIP5 models (Supplementary Note 1). Therefore, the strong TCR in CM3 does not imply that this model's relatively large simulated internal variability should be rejected. While a recent study[27] excluded CM3 from some of their calculations because the model's NINO3.4 SST variability differs by >20% from observed values, in our view the large centennial-scale modulation of tropical Pacific SST variability in long control runs[28] suggests that requiring model-observation agreement within 20% for this type of variability metric is overly restrictive. Further, recent studies[29,30] suggest that CMIP5 models systematically under-estimate regional decadal and interdecadal internal variability, particularly outside of northern high latitudes. These results support using the relatively high variability from CM3 for our analysis.

While internal variability in the real world and models depends to some degree on the base climate state, here we assume that CM3's pre-industrial control run (with no forcing changes) provides a large sample of plausible internal variability applicable

to conditions ranging from the late 1800s through the mid-21st century.

**Potential length of early 21st century slowdown.** Twenty-five large multidecadal internal variability global cooling events in CM3 are identified in Fig. 2a (black segments/circles). The composite average trends of these events and of 25 similar

warming events (Fig. 4) show that during typical strong internal variability events, CM3's surface temperature anomalies are pronounced in the extratropics and high latitudes of both hemispheres. Trend maps for the individual events that were used to form the composites in Fig. 4 are shown in Supplementary Figs 3 and 4, for cooling and warming events, respectively. As discussed in Supplementary Note 2, the spatial structures of temperature trends for the individual events vary considerably

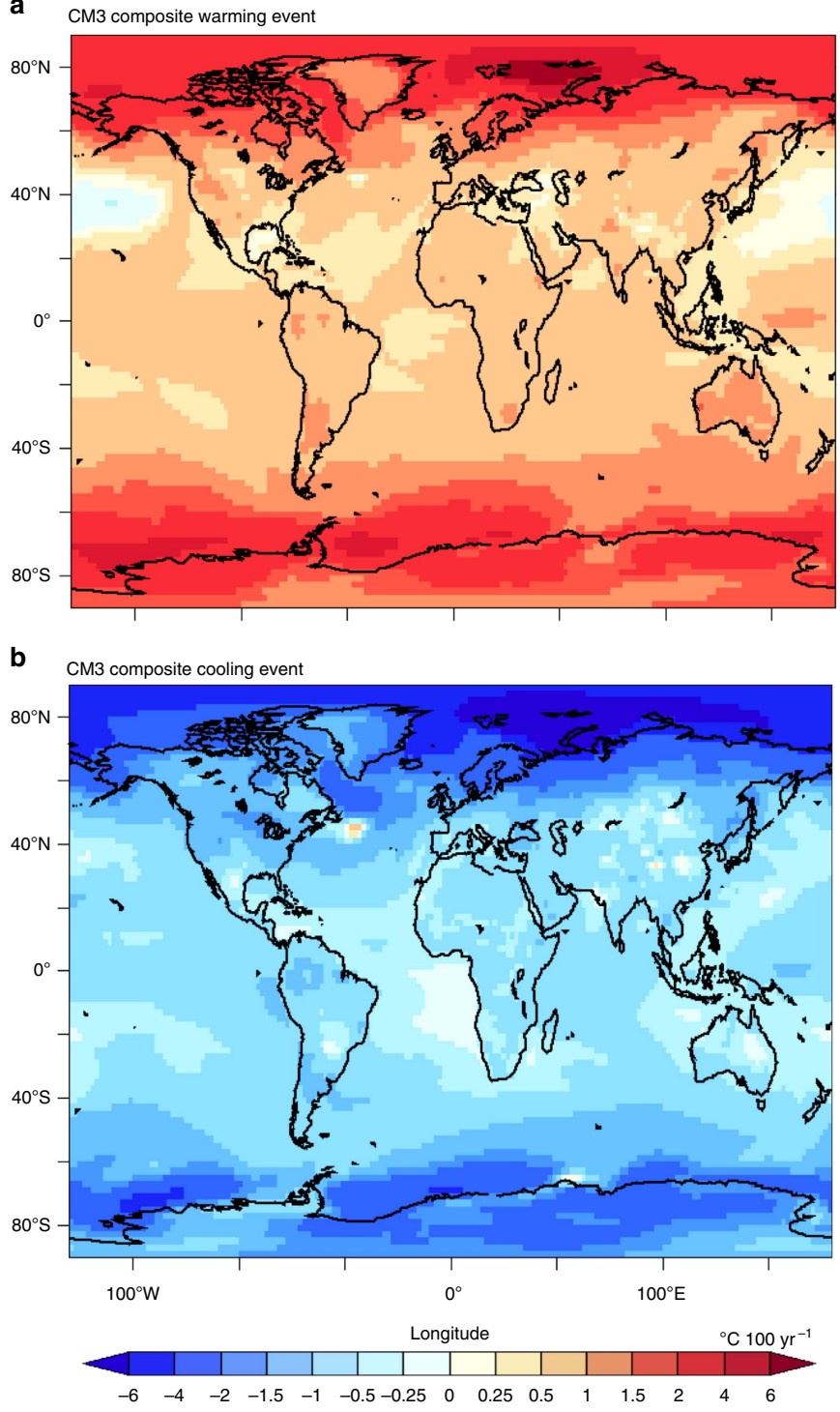

**Figure 4 | Composite trend maps of multidecadal internally generated global warming and cooling events from the CM3 control run.** The composites are based on average of: (**a**) 25 individual warming events shown in Supplementary Fig. 3; (**b**) 25 individual cooling events shown in Supplementary Fig. 4. Unit: °C 100 yr$^{-1}$.

between events, though a consistent feature is a pronounced high-latitude contribution.

The 25 individual cooling events (Supplementary Fig. 3) are each combined with background warming (ensemble mean of CMIP5 Representative Concentration Pathway 8.5 (RCP8.5)

scenarios) to produce a set of synthetic 21st century near-global temperature evolution curves (Fig. 5a; see Methods). These synthetic timeseries suggest that the current slowdown may be prolonged up to about another decade (∼2025) for a few particularly prominent cooling events. However, for most of the

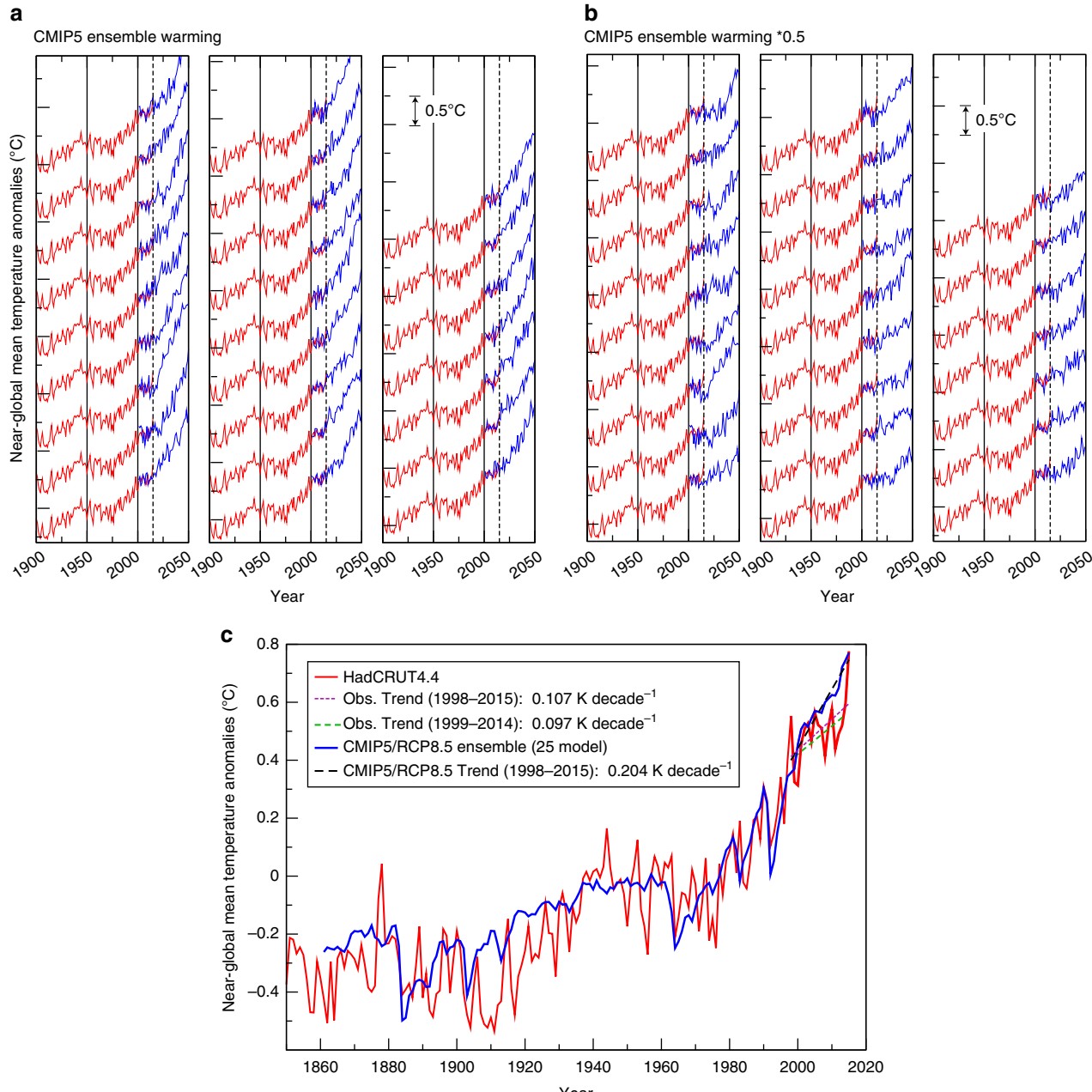

**Figure 5 | Historical global mean temperature timeseries from models and observations along with future model projections. (a)** Synthetic near-global (60°S-80°N) annual time series combining historical observations from HadCRUT4.4 (red) with several projections from climate models (blue). Each of the 25 projections in **a,b** combines the CMIP5 multi-model RCP8.5 warming projection with a different episode of multidecadal internal variability cooling from the CM3 control run. Series are plotted with an arbitrary vertical offset for display purposes (see 0.5 °C scale indicator; tic marks on vertical axis are at 0.5 °C intervals). The dotted vertical lines denote the year 2015 for reference. The synthetic series are constructed using the observed mean anomaly for 1996–2005 as the reference value. Each model projection begins in 2001, using the first year of the CM3 control run segment combined with a long-term warming projection of near-global surface air temperature. The warming projection contribution is adjusted to have zero magnitude in year 2000. The CM3 anomalies, averaged over years 1–10 of the selected segment, are also adjusted to have the observed reference value. (**b**) As in **a** but the long-term warming projection contribution is adjusted by the factor 0.5 to explore effects of a smaller transient climate sensitivity than that simulated in the CMIP5 multi-model ensemble. (**c**) Observed (HadCRUT4.4; red) and CMIP5 multi-model ensemble (All-Forcing/RCP8.5; n = 25) simulated (blue) near global temperature anomalies (80 N-60 S, using modelled SST over ocean regions and 2-metre air temperature over land. Model data masked with observed missing data mask. Anomalies are relative to 1961–1990. HadCRUT4.4 observed (purple dashed: 1998–2015; or green dashed: 1999–2014) and CMIP5 modelled (black dashed: 1998–2015) trends are compared over recent periods (see legend).

25 major CM3 cooling events the hiatus or slowdown appears to end around the present time (dashed vertical lines demark 2015) or even be barely discernible in the synthetic record. We do not include in our analysis the potential effects of future volcanic eruptions[31,32] on our hiatus/slowdown duration statistics.

The synthetic series in Fig. 5a also show examples of greatly accelerated warming lasting a decade or more, which are evidently spring-back effects as an internal variability cooling episode is followed by a strong internal variability warming episode. The strong warming episodes are further amplified by the underlying forced warming trend. One extreme example shows a warming of almost $1\,°C$ in 15 years—a much greater 15-year warming rate than has occurred in the observations to date (red curves). These spring-back warmings illustrate another important potential consequence of strong internal multidecadal variability as simulated in CM3, and reinforce the need to better understand whether such internal variability actually occurs in the real world.

Recent estimates of TCR based on observations ($0.9$–$2.0\,°C$ (ref. 33), $0.90$–$2.50\,°C$ (ref. 34) and $1.0$–$3.3\,°C$ (ref. 35); with medians of 1.3, 1.33 and $1.66\,°C$, respectively) come to differing conclusions on whether the CMIP5 models substantially overestimate the TCR on average (CMIP5 mean of $1.8\,°C$), but generally agree that a low-end estimate for TCR is roughly half that of the CMIP5 models. Accounting for different efficacies of various forcing agents could lead to further upward revision of the observation-based TCR estimates[36]. Here we explore the potential influence of a low-end TCR on the near-term global temperature evolution by reducing the CMIP5 background warming by a factor of 2. The resulting synthetic timeseries (Fig. 5b) show a number of cases with hiatus-like or warming-slowdown behaviour extending well beyond 2015. Several examples of these synthetic series even show an early 21st century hiatus/slowdown extending to be roughly comparable to the 1941–1973 hiatus. Using the CMIP5 RCP4.5 scenario instead of RCP8.5 (not shown) gives similar results to Fig. 5a,b through about 2030, but with less overall warming by 2050, as expected.

Recent near-global temperature trends in observations (HadCRUT4.4, see Methods) and models are compared in Fig. 5c (dashed lines). The warming rate from El Niño peak to peak (1998 to 2015) in observations was about $0.11\,K\,decade^{-1}$, and similar ($0.10\,K\,decade^{-1}$) if the two El Niño years at each end are excluded (that is, 1999 to 2014). In contrast, the background warming rate from external forcing (neglecting missing volcanic forcing[12,16]) over 1998–2015 from the CMIP5 models (All-Forcing historical runs, extended with either RCP4.5

or RCP8.5 scenario) is $0.2\,K\,decade^{-1}$. Reducing the CMIP5 forced warming rate by 50% (to $0.1\,K\,decade^{-1}$) yields a trend close to the observed warming rate. Table 1 indicates that if the CMIP5 models overestimate TCR by a factor of two, we might expect global mean temperatures over the next few decades to continue (on average) along the observed trend trajectories since about 1998 (dashed blue and black lines in Fig. 5c), as indicated by the $\sim 50\%$ probability of continued warming at $0.1\,K\,decade^{-1}$ or less in the bottom row of the table. We refer to this scenario as a continued slowdown of global warming.

Estimated probabilities, based on CM3 internal variability, for various trend scenarios (starting in 1998 and extending through 2030) are summarized in Table 1, assuming two different background warming rates. Conditional probabilities, also in Table 1, focus on scenarios that closely follow the historical trend behaviour since 1998. A hiatus of global warming (defined here as net trend of zero or less, beginning in 1998 and ending in some future year) would be very unlikely (1% probability or less) according to our simulations, if we assume the transient climate sensitivity in the CMIP5 ensemble is correct. If we halve the CMIP5 warming rate, then the probability of a hiatus beginning in 1998 being sustained through 2015, 2020, 2025 and 2030 is estimated as 15%, 9, 5 and 3%, respectively (see examples in Fig. 5b). However, these estimates do not take into account knowledge about the observed warming since 1998. Restricting to only the 500 (of $\sim 5170$ total) overlapped cases derived from the CM3 control run with global net trends over 1998–2015 that are closest to observed ($0.11\,K\,decade^{-1}$), the probability of a hiatus (trend $<0$) extending to 2020, 2025 or 2030 drops to 2% or less, even with the weaker forced warming trend. Thus, the constraint of the observed warming since 1998 greatly reduces the odds of a long hiatus (like that from 1941–1973) now occurring.

A particularly interesting case (Table 1) is the estimated probability, using CM3's variability and the unmodified CMIP5 ensemble background warming rate ($0.2\,K\,decade^{-1}$), that a global warming slowdown (net trend $<0.1\,K\,decade^{-1}$) will still persist through 2020, 2025 or 2030. While these probabilities are relatively low if we consider all CM3 internal variability cases (9, 5 and 3%), the probabilities rise to 16%, 11% and 8%, respectively, if we restrict the calculation to the 500 synthetic cases closest to the observed warming rate during 1998–2015. A previous study[14] has shown that the set of individual CMIP5 ensemble members with lower warming rates ($<0.1\,K\,decade^{-1}$ for at least 14 years) during 1995–2015 have very similar long-term projected warming rates to 2100. Here our analysis suggests that simulating a warming slowdown like that observed from 1998–2015 makes it

**Table 1 | Probabilities of a hiatus or warming slowdown.**

| | 1998–2015 | 1998–2020 | 1998–2025 | 1998–2030 |
|---|---|---|---|---|
| **Hiatus: net trend < 0** | | | | |
| CMIP5 mean (FW = 0.2 K decade$^{-1}$) and IV < − 0.2 K decade$^{-1}$ | 0.01 | <0.01 | <0.01 | <0.01 |
| | | **<0.01** | **<0.01** | **<0.01** |
| CMIP5*0.5 (FW = 0.1 K decade$^{-1}$) and IV < − 0.1 K decade$^{-1}$ | 0.15 | 0.09 | 0.05 | 0.03 |
| | | **<0.01** | **0.02** | **0.02** |
| | | | | |
| **Warming slowdown: net trend < 0.1 K decade$^{-1}$** | | | | |
| CMIP5 mean (FW = 0.2 K decade$^{-1}$) and IV < − 0.1 K decade$^{-1}$ | 0.15 | 0.09 | 0.05 | 0.03 |
| | | **0.16** | **0.11** | **0.08** |
| CMIP5*0.5 (FW = 0.1 K decade$^{-1}$) and IV < 0 K decade$^{-1}$ | 0.51 | 0.51 | 0.51 | 0.52 |
| | | **0.48** | **0.51** | **0.53** |

IV refers to trends generated by internal variability in the GFDL CM3 model. CMIP5 refers to the ensemble mean global warming in a CMIP5 25-model ensemble of All-Forcing Historical simulations extended past 2005 with the RCP4.5 scenario. CMIP5 * 0.5 reduces the modelled warming rate by 50%. FW refers to the forced warming trend, 0.2 K decade$^{-1}$ in CMIP5, 0.1 K decade$^{-1}$ in CMIP5*0.5. A hiatus is defined as net trend (background CMIP5 warming plus CM3 internal variability) of less than zero over the indicated period. A slowdown is defined as net trend $<0.1\,K\,decade^{-1}$ over the indicated period. Non-bold probabilities combine all possible internal variability samples from the CM3 model with the CMIP5 background warming. Bold probabilities use the $\sim 10\%$ of the samples that are closest to the observed trend of 0.107 K decade$^{-1}$ over the 18-year period 1998–2015.

more likely that a continued slowdown will occur over the next decade or so, assuming the background warming rate of CMIP5 models is accurate. In short, if CM3's internal variability is realistic, there is some chance that a rapid underlying warming rate of 0.2 K decade$^{-1}$ could be ongoing as of 2015, but that this warming signal has been substantially masked (and may continue to be masked for even another decade or more) by an internal variability cooling episode.

Another interesting case to consider is that of the low-end TCR (0.1 K decade$^{-1}$), which results in a forced warming rate essentially the same as the recent observed warming trend (magenta and green dashed lines in Fig. 5c). If the global temperatures since 1998 have essentially followed a forced warming-only trajectory, then future internal variability could deviate either up or down from this trajectory and create, for example, a near-term hiatus or accelerated warming period. The analysis in Table 1 concerning the probability of an 18-year hiatus beginning in 1998 would, in such a low-end TCR scenario, provide rough guidance for the case of 2015 as a new start year (that is, implying a 15% chance of an 18-year hiatus beginning around 2015).

**Internal variability contributions to observed trends**. We have shown (Fig. 2) that internal variability generated by CM3 cannot explain the long-term increase in global mean temperatures since the late 1800s. Here we further explore the potential contributions of simulated internal variability to observed multidecadal trends by comparing multidecadal trend maps from GISTEMP observations (Methods) with simulated trends combining CMIP5 All-Forcing historical runs and CM3 control run variability, focusing on much shorter periods of ~15–30 years. The five key epochs of global temperature since 1880 (Fig. 1) are each examined separately (Fig. 6; Methods).

For the 1880–1908 cooling period, the CMIP5 All-Forcing multi-model ensemble forced response (Fig. 6b) shows generally weak positive trends over most regions with relatively little resemblance to the observed trend map (Fig. 6a). However, when this CMIP5 forced response is combined with the ten internal variability events from CM3 that produce the best agreement with observations (Fig. 6c), the combined response matches better the observed 1880–1908 change pattern, reducing the mean absolute deviation with the observed trend by 16% compared with the forced response alone (Fig. 6b). During this period, the CMIP5 ensemble global cooling response to the Krakatau eruption of 1883 (for example, Fig. 5c or ref. 30) was larger than the observed temporary cooling.

Early 20th century warming (1908–1941) extended over most of the globe, except for the far western Pacific near Japan and parts of the deep southern ocean (Fig. 6d). The CMIP5 All-Forcing multi-model mean (Fig. 6e) also shows broad-scale warming over most of the globe, but at a smaller rate than observed. Trend maps combining this CMIP5 forced response with selected CM3 internal variability events (Fig. 6f) are much closer to observed trends than the CMIP5 forced response alone (mean absolute deviation 24% smaller). Including the CM3 internal variability events also results in some slight cooling in the deep southern ocean, similar to—though weaker than—observations. These results support earlier studies[7,8] that tentatively attributed early 20th century global warming to a combination of externally forced warming and internal climate variability.

For the mid 20th century hiatus (1941–1973), an important role for internal climate variability is suggested, as the CMIP5 forced response deviates from the observed spatial pattern. This period featured a slight cooling tendency globally (Fig. 1), lasting >30 years. Observations for this period (Fig. 6g) show pronounced northern hemisphere extratropical cooling along

with several warming and cooling regions in the southern hemisphere. There is not clear evidence in Fig. 6g for a pronounced triangular-shaped Interdecadal Pacific Oscillation-like eastern Pacific cooling pattern like that observed[19] over the recent period 1992–2011. The CMIP5 All-Forcing ensemble (Fig. 6h) depicts a weak broad-scale cooling over most regions except the deep southern hemisphere. The selected CM3 internal variability episodes combined with the CMIP5 forced pattern (Fig. 6i) agree better with observations (mean absolute deviation reduced by 11%), including the pronounced interhemispheric asymmetry (cooling northern hemisphere and warming southern hemisphere). While aerosol forcing potentially could have produced this asymmetry, the CMIP5 ensemble's interhemispheric asymmetry is more muted than observed. The observed pattern thus may reflect a strong northern hemispheric cooling/southern hemisphere warming from internal variability, together with a broad-scale minor cooling due to radiative forcing changes. Remaining uncertainties in climate forcing agents and in the models' responses to those forcings[1] preclude a more confident assessment.

For the late 21st century rapid warming period (1973–2001), observed trends were similar in some respects to those for the early 20th century warming (cf. Fig. 6d,j). Maps for both periods featured broad-scale warming patterns covering most land and ocean regions, excepting parts of the deep southern hemisphere. The regional cooling in the deep southern hemisphere in the 1973–2001 GISTEMP trend map (Fig. 6j) is not present in the CMIP5 All-Forcing ensemble-mean trend map (Fig. 6k), but can be partly captured by including selected internal variability episodes from CM3 (Fig. 6l). Ensemble-mean trends for this period from some individual CMIP5 models (not shown) depict regional cooling features in the deep southern hemisphere similar to observations, so it is not yet clear whether these regional features were caused by internal variability or external forcing. The CMIP5 ensemble global-mean trend is close to observed (Fig. 1). Nonetheless, if CMIP5 models overestimate the TCR (forced warming rate), internal climate variability may have also played a significant role in the late 20th century global warming.

For the early 21st century slowdown period (2001–2014), the faster-than-observed externally forced warming in CMIP5 models supports a role for internal variability-induced cooling in recent years[11,13–15,20,23,30,37]. The recent (2001–2014) observations (Fig. 6m) feature a mixture of cooling and warming regional trends with some strong cooling in the eastern Pacific, differing clearly from the CMIP5 forced response (Fig. 6n). We used 2014 as the end year to avoid including the highly anomalous El Niño year of 2015 in the short-term trend. The addition of that single year would alter the observed trend map substantially (not shown) relative to the map for 2001–2014, introducing weaker, less-pronounced cooling trends throughout much of the eastern tropical and subtropical Pacific. The lack of agreement between simulated and observed trends is not surprising given the relatively short period analysed (14 years). Excepting major volcanic periods, such relatively short trends are typically more likely to include larger contributions from internal variability versus forced change. Including the selected sample of CM3 modelled internal variability (Fig. 6o) results in a trend map more closely resembling observations (mean absolute deviation 19% smaller), further suggesting that internal variability can help provide a more plausible explanation for observed changes during 2001–2014 compared with external forcing alone (Fig. 6n).

As an alternative approach, we compare CM3's internal variability alone to observed trend maps over each of the five epochs discussed above (Fig. 7), to explore how well internal variability trends can reproduce the observed trends (a, d, g, j and m) assuming no externally forced contributions. That is, the

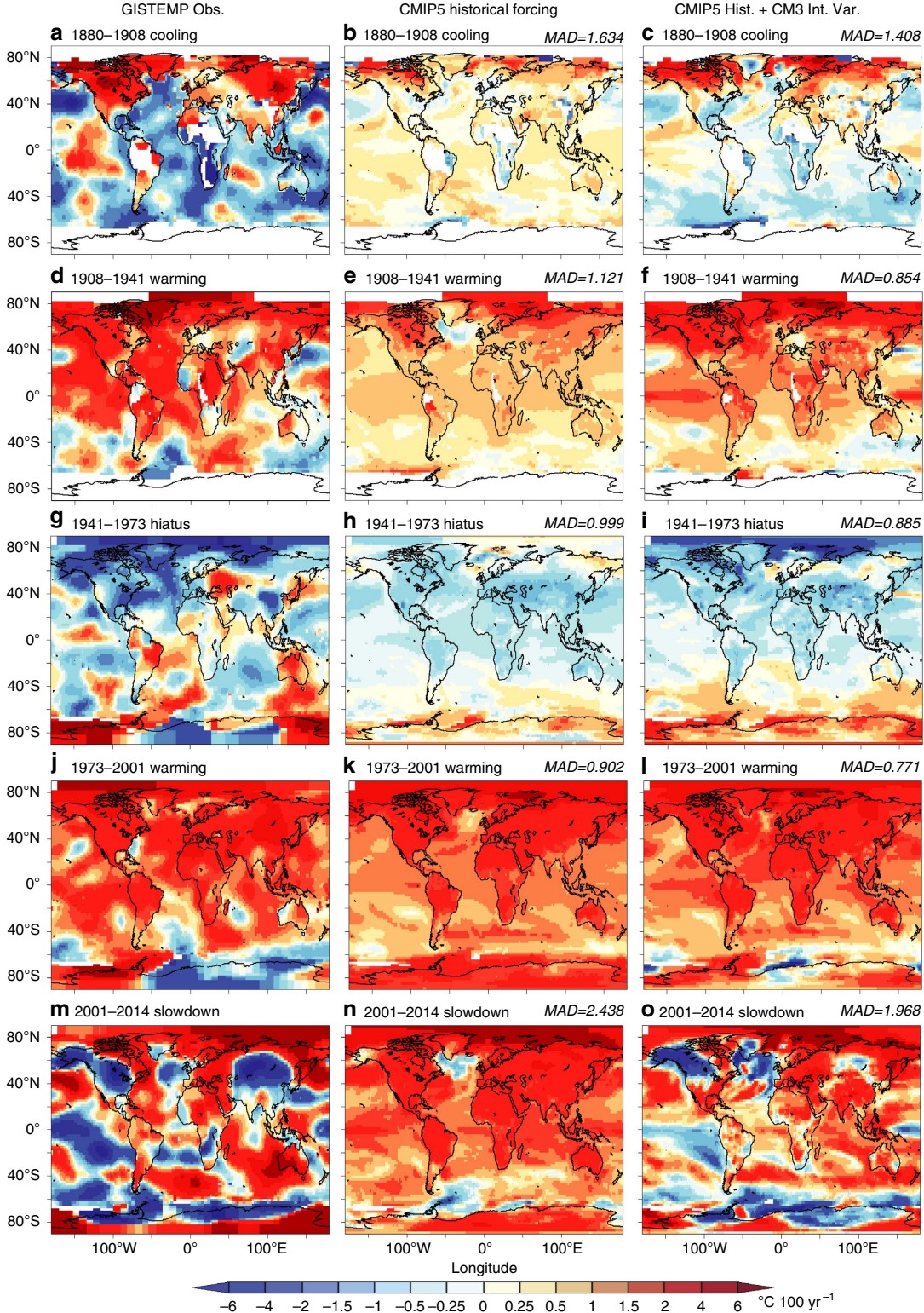

**Figure 6 | Maps of observed and simulated surface temperature trends for five epochs.** GISTEMP observed trends (**a,d,g,j** and **m**); CMIP5 ensemble All-Forcing trends (**b,e,h,k** and **n**); and CMIP5 All-Forcing ensemble plus CM3 control run trends based on selected multidecadal cooling/warming events in CM3 (**c,f,i,l** and **o**). For (**c,f,i,l** and **o**), the 10 CM3 control run events that result in the smallest globally averaged mean absolute deviation (MAD) of the trend map compared with the observed trends (**a,d,g,j** and **m**) were composited. The MAD of each modelled map versus the observed trend map is listed above each simulation panel (unit: °C 100 yr$^{-1}$) for the CMIP5 historical forcing only and for the CMIP5 trend combined with CM3 internal variability. The five epochs (year ranges) examined include: (**a–c**) Late 1880s cooling (1880–1908); (**d–f**) Early 20th century warming (1908–1941); (**g–i**) Mid-20th century hiatus (1941–1973); (**j–l**) Late 20th century warming (1973–2001); and (**m–o**) Early 21st century warming slowdown (2001–2014). Model results are based on combined SST/surface air temperature data, with the observed missing data mask applied to model data. Grid boxes where <33% of months have observed data available for the trend period were masked out of the trend maps (denoted by white shading).

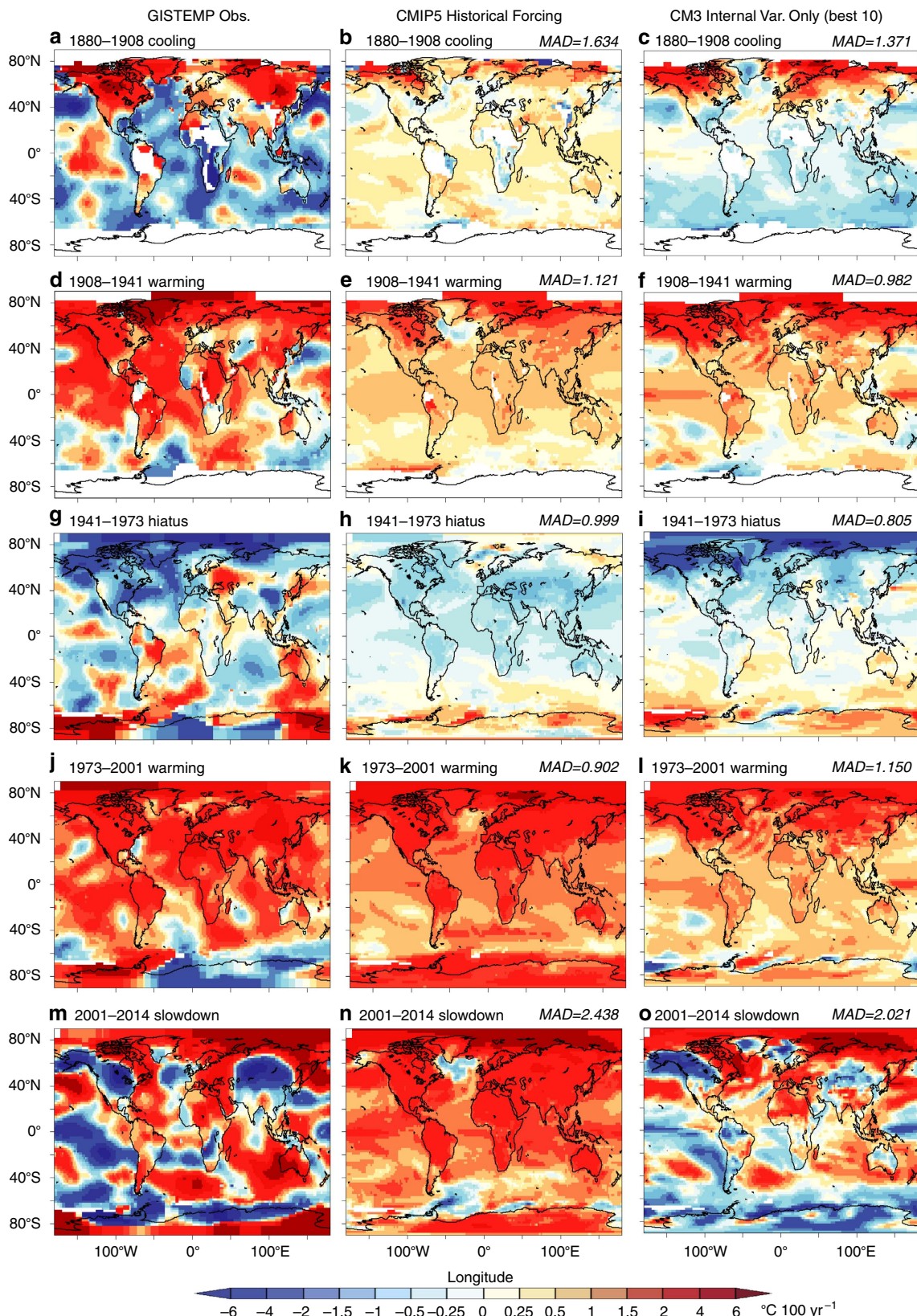

**Figure 7 | Maps comparing historical observed and CMIP5-simulated trends with trends based on CM3 internal variability only.** Panels (**a,b,d,e,g,h,j,k,m** and **n**) are repeated from Fig. 6, while panels (**c,f,i,l** and **o**) are based on CM3 model-simulated internal variability only. Unit: °C 100 yr$^{-1}$.

CMIP5 historical run trends contributions are assumed to be zero. For each epoch analysed, the ten periods from the CM3 control run having the lowest global mean absolute deviation from the observed trends are averaged to create the control run composite trend maps. The results (Fig. 7, panels c,f,i,l and o) show that CM3 internal variability alone can reproduce the

observed trends fairly well during hiatus or slowdown periods (1880–1908; 1941–1973; and 2001–2014), without invoking any radiative forcing-produced trends. This is shown by comparing mean absolute deviation values of Fig. 7 panels (c,f,i,l and o) to those from Fig. 6 (panels c,f,i,l and o), which include both historical forcing and internal variability contributions. However, CM3's simulated internal variability alone appears unable to reproduce a global scale warming trend nearly as strong as observed during the 1908–1941 and 1973–2001 rapid warming periods. For these periods, some warming contributions from external forcing seem crucial for explaining the pronounced observed trends, reinforcing the importance[1] of external forcing (particularly anthropogenic forcing) for explaining much of observed long-term warming since the late 1800s and since the mid 20th century. Maps showing two alternative estimates of the internal variability contribution to the observed trend maps (Supplementary Fig. 5) are discussed in Supplementary Note 3.

## Discussion

We investigated the potential influence of internal multidecadal climate variability on near-global mean surface temperature evolution over the past century and next several decades using the GFDL CM3 model, which exhibits the strongest multidecadal variability among the CMIP5 models. We explored whether CM3-like internal climate variability could, in the coming decade or so, produce a prolonged warming slowdown or hiatus. We also noted some simulated cases of rapid global warming lasting a decade or more as an internal cooling episode flips to a warming phase and combines with the underlying long-term forced warming trend. If the slowdown in warming over the recent period (for example, 1998–2015, or 1999–2014) turns out to have resulted mostly from Pacific-centric internal variability, for example, the Interdecadal Pacific Oscillation or Pacific Decadal Oscillation[2,15,19,20,23], combined with background warming like that in the CMIP5 ensemble, these variations might likely have a shorter timescale (∼10–15 years of cooling) than many of the longer timescale (∼30 years) internal variability events from CM3 highlighted in our analysis. In that case, we would speculate that—if the TCR of the climate system is close to that of the CMIP5 ensemble mean—the current global warming slowdown will end soon or perhaps has already ended, given the strong positive anomaly in global mean temperatures in 2015. However, our analysis suggests that multidecadal internal variability involving the Atlantic, and particularly higher latitude regions, could potentially lead to a more prolonged influence on global temperature than is thought to have occurred to date. This longer timescale internal multidecadal variability in CM3 has strong regional expressions, particularly in high latitudes, and is not dominated by variability in the tropical Pacific Ocean. Our synthetic timeseries suggest that it is possible for an early 21st century warming hiatus or slowdown to extend past 2030, and in a few extreme cases eventually even come to resemble the 1941–1973 hiatus. The most extreme cases could result from a large internal variability cooling event commencing around year 2000, together with a weaker warming trend than projected by the CMIP5 model ensemble mean, due to either a lower TCR[33–35] or a smaller future positive forcing. However, according to the model simulations, the probability of a hiatus (zero or negative trend) extending from 1998 to some year beyond 2015 is now very small (2%), even if the CMIP5 models have warming rates that are a factor of two too large.

If the transient warming rates of the CMIP5 multimodel ensemble turn out to be a factor of two too large, in line with the low-end estimates of recent studies[33–35], then the recent observed near-global warming rate of about 0.1 K decade$^{-1}$ would be

representative of what to expect, on average, over the next few decades, in contrast to the 0.2 K decade$^{-1}$ background warming projected by the CMIP5 models. Even if the ensemble transient warming rate from CMIP5 models (0.2 K decade$^{-1}$) turns out to be correct, we find that strong internal variability could still extend the recent warming slowdown (<0.1 K decade$^{-1}$) for another decade or more, at least partially masking the strong underlying warming trend. Specifically, in this case, we estimate that a warming slowdown (<0.1 K decade$^{-1}$ net trend beginning in 1998) could be sustained by a strong cooling phase of internal variability through 2020, 2025 or 2030 with probabilities 16%, 11% and 8%, respectively, assuming CM3's internal variability is realistic.

Our analysis suggesting a potential extension of an early 21st century slowdown to ∼2030 assumes no substantial volcanic forcing or unusually weak solar forcing over the period. If a climatically important eruption(s) occurred[31,32], or if the sun transitioned to Maunder-Minimum-like solar conditions[38–40], this would likely further prolong/deepen any ongoing hiatus or slowdown and perhaps lead to a subsequent warming recovery surge[31]. CMIP5 model simulations suggest the Mt Agung eruption may have played such a role[30] in prolonging the mid-20th century hiatus. Maher et al.[32] conclude that hiatus decades will be very unlikely to occur beyond 2030 for high greenhouse emissions scenarios in CMIP5 models (for example, RCP8.5) even including volcanic eruptions, although hiatuses in the late 21st century remain relatively likely under scenarios with lower externally forced warming rates (for example, RCP2.6).

We have further explored the possible contributions of internal variability to historical surface temperature changes. Our analysis of global mean temperatures and trend maps across five historical epochs suggests that internal variability may have contributed significantly, along with external forcing, to several prominent warming and hiatus epochs since the late 1800s. However, confident assessment of the relative role of external forcing (including aerosols) and internal variability in these historical changes remains a significant challenge at global and regional scales[41–43]. Strong multidecadal internal variability events, such as simulated in the 5200-year CM3 control, may have contributed to a number of such changes. Uncertainties in aerosol forcing and the climate system response to this forcing[37,44], in the potential magnitude of internal climate variability[29,30,45], and the relatively short observational record for assessing internal variability all contribute to this research challenge and related uncertainties in future projections[46,47].

The evolution of global mean temperature and the recent slowdown in global warming pose interesting opportunities to test our understanding of the climate system, and determine strengths and weaknesses of current models, particularly since models cannot be tuned to observations that have not yet occurred. Our analysis of model simulations suggest some conditions whereby an early 21st century global warming slowdown could potentially last much longer (to ∼2030) than is generally expected, as well as scenarios where a rapid temporary acceleration of warming might occur. The possibility of such behaviour in the climate system in coming decades, while not a prediction, should motivate more study of internal multidecadal climate variability and its mechanisms.

## Methods

**CMIP5 ensemble historical run timeseries.** The CMIP5 multi-model ensemble historical run timeseries (Figs 1 and 5c) is formed from the 25 models and ensemble members identified in Supplementary Fig. 1. Model data (blended SST and surface air temperature data, using skin temperature for sea ice-covered regions) is masked out where and when data are missing in the HadCRUT4 data set[30]. Control run drift effects (Supplementary Fig. 1) on the historical run solutions are reduced by subtracting the linear trend field in each control run

(computed for years 1–165) from each historical run ensemble member for that model. For FGOALS-g2 the control run drift adjustment is computed using years 151–315. The model ensemble historical series in Figs 1 and 5c have been extended beyond 2005 using RCP8.5 projection scenarios. Ensemble results for 2006 to 2015 are similar using the RCP4.5 scenario.

**Model control run variability.** A 5000-year record of global mean surface air temperature from a long pre-industrial control simulation of the GFDL CM3 climate model[48,49] is shown in Fig. 2. The near global (80°N-60°S) record is shown in the top panel, while the northern hemisphere extratropics (23.6°N-90°N) is shown in the bottom panel. We restrict our analysis to 80°N-60°S (rather than global) for model and observations due to the relatively sparse observations poleward of this domain in both hemispheres. Also shown on the plots for comparison (arbitrary vertical offset) are the observed mean surface temperature anomaly series (HadCRUT4 data[50], also 80°N-60°S or 23.6°N-90°N, but combining SST over ocean with surface air temperature over land), the CMIP5 residual created by subtracting the CMIP5 multi-model ensemble forced response from the HadCRUT4 observations, and a linear trend residual obtained by removing a linear trend from the HadCRUT4. During the first ~1,500 years of the control run (top curve of the pair of CM3 control model curves on each plot) the model drifts slowly towards a warmer climate state, then settles into a quasi-equilibrium state in terms of global and NH extratropical mean temperature. We fit separate linear trends through years 1–1530 and years 1531–5200 of the control run data to remove most of the influence of this secular drift from our subsequent global temperature evolution analysis.

**Synthetic global mean time series.** We identify 25 CM3 control run variability cooling episodes, which are combined with observed historical near-global (80°N-60°S) temperature and 21st century forced change from the multi-model ensemble of CMIP5 models (RCP8.5 scenario, with RCP4.5 also used as a sensitivity test), to create 25 synthetic global temperature annual-mean timeseries. For these prototype events, we assume that the internal-variability driven temperature anomaly peaks in year 2000, so that the 10-year mean corresponding to the warm peak value at the beginning of the cooling period is set equal to the observed anomaly averaged over 1996–2005. Beginning in 2001, a long-term, externally forced warming from the CMIP5 multi-model ensemble is added to each of the CM3 control run cooling episodes, adjusted so that the warming contribution is zero for the year 2000. We thus produce 25 synthetic future evolutions of global temperature anomalies under the assumption of a very strong global cooling episode resulting from internal variability, as simulated in the CM3 model, commencing around the year 2000.

**Analysis of 20th century epochs.** For mapped analysis of the spatial structure and magnitude of trends during five key historical epochs, we have chosen to emphasize the GISTEMP data set[51] rather than the HadCRUT4 data set, since the GISTEMP has better spatial coverage in the high-latitude regions. This better spatial coverage is produced by imputing values for regions with missing data based on extrapolation from nearby regions with available data—that is, using all grid boxes located with 1,200 km of a station with available data. While trends in the extrapolated regions should be treated with caution, here we have chosen to use the GISTEMP trend maps for a preliminary exploration of the trend patterns, particularly since an important component of the modelled internal variability in CM3 extends into data-sparse regions of high latitudes in both hemispheres, as we show in our analysis.

The combined CMIP5 forced response and CM3 internal variability trend maps (for example, Fig. 6c,f,i,l and o) were created by adding the CMIP5 All-Forcing ensemble trend to all possible segments of that length from the CM3 control run to create a large sample of ~5,000 possible trend maps. We compare this large sample with the observed trend and identify the 10 independent (non-overlapping) members of the sample that have the lowest global mean absolute deviation from the observed trend field. The average of these best-10 samples is shown for each epoch in Fig. 6 panels (c,f,i,l and o) and (for the case where the CMIP5 All-Forcing contribution is not included) in Fig. 7 panels (c,f,i,l and o).

**Data availability.** The 5,200-year modelled surface temperature data set from the CM3 model control and/or the source code for the CM3 model are available from the authors upon request. Other model data is available from the CMIP5 data portal (http://cmip-pcmdi.llnl.gov/cmip5/data_portal.html). Sources for the observed data sets are: HadCRUT4 (ref. 50): http://www.metoffice.gov.uk/hadobs/hadcrut4/; GISTEMP[51] (data accessed 2016-02-19): http://data.giss.nasa.gov/gistemp/; and NOAA[2] (accessed on 2016-08-11): http://www.ncdc.noaa.gov/monitoring-references/faq/anomalies.php.

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

## Acknowledgements

We acknowledge PCMDI and the CMIP5 modelling groups for model data, and NASA/GISS, UKMO Hadley Centre, UEA Climatic Research Unit and NOAA National Centers for Environmental Information for observational data. We thank three external reviewers for comments that improved the study.

## Author contributions

T.K. formulated the study, performed analysis and wrote the paper. R.Z. contributed towards formulation of the analysis. L.H. contributed the 5200 year CM3 control run integration. R.Z. and L.H. assisted in writing the paper.

## Additional information

**Competing financial interests:** The authors declare no competing financial interests.

