## [Peer Review File · Nature Communications]

Reviewers' comments:

Reviewer #1 (Remarks to the Author):

This is one of the most important papers that I've read in awhile. For the first time, I see a plausible interpretation of the global temperature variations for the past ~150 years, that realistically accounts for the effects of internal variability. The issue of multidecadal climate variability in the evolution of 20th and 21st century climate change is extremely important yet difficult to get a handle on, as it is very difficult to deconvolute the signals of forced from internal variability.

The elements of the study that are particularly meritorious:

- use of a model (CM3) that has high amplitude and relatively realistic internal variability
- 5000 yr model run to interpret internal variability
- acknowledgment that climate models may be oversensitive to CO₂, and adjusting for this by scaling to lower values of TCR that are empirically derived

The conclusions drawn are appropriate based upon the research, and they appropriately acknowledge the uncertainties. This study presents a new and very useful methodology to test our understanding of the climate system and assess the strengths and weaknesses of climate models.

Reviewer #2 (Remarks to the Author):

This is a fairly interesting and reasonably well-argued paper. It adds to similar papers (below) by contextualizing the recent slowdown in terms of the historical record of decadal variability. And, I appreciated the care taken to put the CM3 simulated magnitude of internal variability into the context of the magnitude of the observed record of decadal variability (and relative to the CMIP5 model simulations as a group). The authors point about not over-interpreting the magnitude of the observed record of decadal variability (given their long control simulation) is a good one.

All of this said, I cannot recommend that this manuscript be published in its present form. The manuscript misses citing and discussing several key papers on the topic. The papers that I have in mind are listed below (and, no doubt there are more). The figures in this manuscript are somewhat poorly selected and generally below standard (Fig. 2, for example needs a magnifying glass to be appreciated), and overall the presentation is far too descriptive for my taste. It's one thing to say that an extended slowdown is possible (in the context of the CM3 control simulation) but its quite another thing to establish the probability of such events (with proper uncertainties).

1. Meehl et al., Nature Clim. Change 4, 898-902 (2014)
2. England et al. Nature Clim. Change 5, 394-396 (2015)
3. Roberts et al. Nature Clim. Change 5, 337-342 (2015)
4. Risbet et al. Nature Clim. Change 4, 835-840 (2014)
5. Schurer, et al. Geophys. Res. Lett. 5974-5982 (2015)

Reviewer #3 (Remarks to the Author):

Review for Nature Communications

"Could a prolonged hiatus or slowdown in global warming occur in the early 21st century?"

By Knutson, Zhang, and Horowitz

Recommendation: accept subject to major revisions

This paper addresses the issue of possible length of global warming slowdown periods based on a long control run with the GFDL model. The topic of internally generated decadal climate variability is an interesting one, and this paper shows that quite long intervals of global warming slowdown can occur in the GFDL model. The authors note that this is, of course, a model result and therefore only instructive to what may happen. They also address unique features of their model (i.e. high TCR and ECS) that may be producing the magnitude of these results. They then attempt to scale future climate simulations with the decadal variability characteristics of the control run to illustrate the interaction between internally generated variability and externally forced response in producing possible future slowdowns or accelerations.

As such, this paper addresses an important and relevant topic. I have some comments below regarding details of their analyses that the authors may want to consider as they revise the paper.

Detailed comments

1. Lines 40-43: Three generations of IPCC assessments (2000, 2007, 2013, based on many studies) have concluded that the 1940s to 70s hiatus was likely mostly due to externally forced response, and this should be mentioned here.
2. Line 52: should be "negative phase of the Interdecadal Pacific Oscillation"
3. Line 96: should be "The composite average trends of these events"
4. Lines 121-133: The duration of hiatus periods in future climate change simulations depends on the magnitude of the positive forcing in the scenario considered. Thus it has been shown that RCP8.5 has virtually no hiatus periods in the CMIP5 multi-model ensemble, while RCP2.6 has many hiatus periods, and this should be mentioned here (Maher, N., A. Sen Gupta, and M. H. England, 2014: Drivers of decadal hiatus periods in the 20th and 21st centuries, *Geophys. Res. Lett.*, 41, doi:10.1002/2014GL060527). In that same vein, it has also been noted for the CMIP5 ensemble that uses 20th century and early 21st century simulations for RCP.5, that at least one ensemble member simulated a slowdown that matched the observed reduced rate of global warming from 2000-2013 that lasted until 2018, a slowdown of 19 years synched to the present time frame (Meehl, G.A., H. Teng and J.M. Arblaster, 2014: Climate model simulations of the observed early-2000s hiatus of global warming. *Nature Climate Change*, 4, 898-902, DOI: 10.1038/NCLIMATE2357).
5. Line 151: Is there any way to quantify "much better"? Also, what was the role of the huge Krakatau eruption in this time period?
6. Line 158: Any way to better quantify "much closer"?
7. Lines 170-171: Some confusing wording here, "...evidence in the observations...as observed"
8. Line 174: Any way to better quantify "much better"?
9. Line 177-181: See comment 1 above
10. Line 198 and later: the use of 2001-2015 probably crosses decadal regimes since it is likely the IPO transitioned in 2014 and choice of start and end dates is important (see Fyfe et al 2016). It may be better to use 2001-2013, or at least 2001-2014. Then in lines 204-206, the authors note that adding 2015 changes the trend map "substantially" compared to a map where the end date is 2014. Indeed, if the IPO transitioned in 2014, excluding 2015 would give a different

answer!

11. Line 209: Any way to better quantify "more closely"

12. Line 227: Here the authors refer to 2001-2014 (see comment 10 above). As noted, it would be better to use either 2001-2013 or 2001-2014 for reasons given in Fyfe et al 2016.

13. Lines 232-233: Yes, likely already ended-see comment 10 above

14. Line 274: "unexpected behavior" is an odd term-the authors have just outlined that either the slowdown will continue or it won't, so we can expect one or the other, and so there is really nothing unexpected about what is happening, we just don't know which of the two alternatives is happening.

Reviewers' comments: (with author responses in red)

Reviewer #1 (Remarks to the Author):

This is one of the most important papers that I've read in awhile. For the first time, I see a plausible interpretation of the global temperature variations for the past ~150 years, that realistically accounts for the effects of internal variability. The issue of multidecadal climate variability in the evolution of 20th and 21st century climate change is extremely important yet difficult to get a handle on, as it is very difficult to deconvolute the signals of forced from internal variability.

The elements of the study that are particularly meritorious:

- use of a model (CM3) that has high amplitude and relatively realistic internal variability
- 5000 yr model run to interpret internal variability
- acknowledgment that climate models may be oversensitive to CO₂, and adjusting for this by scaling to lower values of TCR that are empirically derived

The conclusions drawn are appropriate based upon the research, and they appropriately acknowledge the uncertainties. This study presents a new and very useful methodology to test our understanding of the climate system and assess the strengths and weaknesses of climate models.

There is no response needed for this comment.

Reviewer #2 (Remarks to the Author):

This is a fairly interesting and reasonably well-argued paper. It adds to similar papers (below) by contextualizing the recent slowdown in terms of the historical record of decadal variability. And, I appreciated the care taken to put the CM3 simulated magnitude of internal variability into the context of the magnitude of the observed record of decadal variability (and relative to the CMIP5 model simulations as a group). The authors point about not over-interpreting the magnitude of the observed record of decadal variability (given their long control simulation) is a good one.

All of this said, I cannot recommend that this manuscript be published in its present form. The manuscript misses citing and discussing several key papers on the topic. The papers that I have in mind are listed below (and, no doubt there are more). The figures in this manuscript are somewhat poorly selected and generally below standard (Fig. 2, for example needs a magnifying glass to be appreciated), and overall the presentation is far too descriptive for my taste. It's one thing to say that an extended slowdown is possible (in the context of the CM3 control simulation) but its quite another thing to establish the probability of such events (with proper uncertainties).

1. Meehl et al., Nature Clim. Change 4, 898-902 (2014)
2. England et al. Nature Clim. Change 5, 394-396 (2015)
3. Roberts et al. Nature Clim. Change 5, 337-342 (2015)
4. Risbet et al. Nature Clim. Change 4, 835-840 (2014)
5. Schurer, et al. Geophys. Res. Lett. 5974-5982 (2015)

Response:

We thank the reviewer for these comments which have led us to revise and build upon certain aspects of our analysis, resulting in what we think is a better manuscript overall. For example, our abstract has been substantially rewritten, as we have gone from stating that certain events are possible in the context of CM3 to estimating the probabilities of certain types of trend outcomes (again based on CM3 internal variability) in addressing the reviewer's comments.

We have added the 5 additional suggested references which the reviewer pointed out. Roberts et al. is a particularly relevant reference for the topic of estimating probabilities, and conditional probabilities, of certain trend events, which we found to be a useful starting point in addressing the main comment of the reviewer. In comparison to Roberts et al., we look at the impact of the recent very warm year (2015) which lowers the odds now of having an extended hiatus (zero trend) beginning in 1998, because now we have had some substantial warming at about 0.1 K/decade since 1998. We use CM3 control variability, which Roberts et al. exclude because the NINO3 variability differs by more than 20% from the observed. We consider their approach to model screening, while well-intended, to be overly restrictive, given the large multidecadal to centennial scale modulations of internal ENSO variability in long control runs shown by Wittenberg and others. Such modulations (which we illustrate in our supplemental material Fig, S2 for the case of near global temperature anomalies in CM3) result in periods of close agreement and pronounced disagreement with a given observed target variability based on only 140 years or record, for example, and during periods of disagreement the differences can be much greater than 20%. We examine the impact of possible lower TCR in a clearer and more apparent way than in Roberts et al. and we look at probabilities of a continued slowdown (trend of 0.1 K/decade or less) in addition to full hiatus (trend < 0).

We appreciate that the reviewer has trouble seeing fine-scale details in Fig. 2 of the submitted manuscript. The figure panels are much clearer in the postscript panels that we have developed, but once they are imported into the Word document the quality decreases. This can be addressed in the

production phase. In addition, we have reduced the number of such panels in Fig. 4 from 4 to 2, have increased the size of panels a and b modestly, and have a new panel 4c, which shows details such as the 2015 observed anomaly much more clearly. If one tries to identify the 2015 spike in the upper panels (Fig. 4 a, b) one does need to squint, but this issue is not so important now that we are showing the 2015 anomaly more clearly in the new panel c. Panels (a,b) are intended more to illustrate the general trend behavior in different internal variability scenarios (e.g., occurrences of general features like the pronounced 1940-1970 hiatus) than about finer details like the 2015 observed spike.

To address the reviewer's concern about the paper being too descriptive without estimating probabilities (with proper uncertainties), we have developed Table 1 and related text in the latter part of section 3, where we estimate the probabilities of different trend scenarios, at least within the setting of the CM3 internal variability simulation (5200 year sample). We test the influence of different assumptions about background warming rates (0.2 K/decade as in CMIP5 multimodel mean) and $\frac{1}{2}$ that rate, as suggested by recent studies attempting to constrain the TCR from observations. We also explore different trend periods (out to 2030) and different net trend scenarios (zero or negative trend – a hiatus; or continuation of a slowdown or 0.1 K/decade trend). We also estimate conditional probabilities that take into account the knowledge that we already have about observed trend behavior since 1998, and in selecting model scenarios that approximate this observed behavior we refine the probabilities of certain outcomes. In summary, we now have developed estimated probabilities and conditional probabilities, of certain trend outcomes. We have not, however, developed formal confidence intervals for these. Given the uncertainties in global mean temperature internal variability (for which development of truly proper confidence intervals would be a complete research project on its own!); the uncertainties in background warming rates due to TCR uncertainties and forcing uncertainties) this would be a very big task and beyond the scope of the present study, and to our knowledge has not been done to date.

Reviewer #3 (Remarks to the Author):

Review for Nature Communications

"Could a prolonged hiatus or slowdown in global warming occur in the early 21st century?"

By Knutson, Zhang, and Horowitz

Recommendation: accept subject to major revisions

This paper addresses the issue of possible length of global warming slowdown periods based on a long control run with the GFDL model. The topic of internally generated decadal climate variability is an interesting one, and this paper shows that quite long intervals of global warming slowdown can occur in the GFDL model. The authors note that this is, of course, a model result and therefore only instructive to what may happen. They also address unique features of their model (i.e. high TCR and ECS) that may be producing the magnitude of these results. They then attempt to scale future climate simulations with the decadal variability characteristics of the control run to illustrate the interaction between internally generated variability and externally forced response in producing possible future slowdowns or accelerations.

As such, this paper addresses an important and relevant topic. I have some comments below regarding details of their analyses that the authors may want to consider as they revise the paper.

Detailed comments

1. Lines 40-43: Three generations of IPCC assessments (2000, 2007, 2013, based on many studies) have concluded that the 1940s to 70s hiatus was likely mostly due to externally forced response, and this should be mentioned here.

We have added a mention of this in the revision. By the way, I have not noticed anywhere in the IPCC reports where they actually claim this to be the case, but certainly the simulations from the CMIP models used in the IPCC reports, and related IPCC figures of observed vs. simulated global temperature, etc. support such an interpretation.

2. Line 52: should be "negative phase of the Interdecadal Pacific Oscillation"

We have adjusted this text.

3. Line 96: should be "The composite average trends of these events"

We have adjusted the text.

4. Lines 121-133: The duration of hiatus periods in future climate change simulations depends on the magnitude of the positive forcing in the scenario considered. Thus it has been shown that RCP8.5 has virtually no hiatus periods in the CMIP5 multi-model ensemble, while RCP2.6 has many hiatus periods, and this should be mentioned here (Maher, N., A. Sen Gupta, and M. H. England, 2014: Drivers of decadal hiatus periods in the 20th and 21st centuries, *Geophys. Res. Lett.*, 41, doi:10.1002/2014GL060527). In that same vein, it has also been noted for the CMIP5 ensemble that uses 20th century and early 21st century simulations for RCP.5, that at least one ensemble member simulated a slowdown that matched the observed reduced rate of global warming from 2000-2013 that lasted until 2018, a slowdown of 19 years synched to the present time frame (Meehl, G.A., H. Teng and J.M. Arblaster, 2014: Climate model simulations of the observed early-2000s hiatus of global warming. *Nature Climate*

Change, 4, 898-902, DOI: 10.1038/NCLIMATE2357).

We have added these two references.

5. Line 151: Is there any way to quantify "much better"? Also, what was the role of the huge Krakatau eruption in this time period?

We now quantify what we mean by "much better" here and elsewhere in related statements by computing the global mean absolute deviation between observed and simulated trend maps for each panel in Fig. 5 (center and right columns) and for Fig. S6 in supplemental material. These allow us to make more quantitative statements about how much closer to the observed trend one can get by using, for example, a combination of a CMIP5 historical forcing response the internal variability from CM3, or (as in Fig. S6) by using the CM3 internal variability alone.

6. Line 158: Any way to better quantify "much closer"?

See response to comment 5.

7. Lines 170-171: Some confusing wording here, "...evidence in the observations...as observed"

We have adjusted the wording to be less confusing.

8. Line 174: Any way to better quantify "much better"?

See response to comment 5.

9. Line 177-181: See comment 1 above

We did reference the previous IPCC reports (and previous CMIP model historical runs) in addressing comment 1. We don't think we need to repeat that here.

10. Line 198 and later: the use of 2001-2015 probably crosses decadal regimes since it is likely the IPO transitioned in 2014 and choice of start and end dates is important (see Fyfe et al 2016). It may be better to use 2001-2013, or at least 2001-2014. Then in lines 204-206, the authors note that adding 2015 changes the trend map "substantially" compared to a map where the end date is 2014. Indeed, if the IPO transitioned in 2014, excluding 2015 would give a different answer!

We have adjusted figure 5 (m,n,o) and related supplemental figures S6 and S7 to use 2001-2014.

11. Line 209: Any way to better quantify "more closely"

See response to comment 5.

12. Line 227: Here the authors refer to 2001-2014 (see comment 10 above). As noted, it would be better to use either 2001-2013 or 2001-2014 for reasons given in Fyfe et al 2016.

We use 2001-2014 here and now also in Fig. 5 as mentioned in response to comment 10.

13. Lines 232-233: Yes, likely already ended-see comment 10 above

Our new calculations in response to reviewer 2 allow us to estimate, within the context of CM3's internal variability and various assumptions about background warming rates due to radiative forcing response, the probabilities of extended hiatus (zero trend--now probability is low given 2015 record temperatures) or of an extended slowdown of 0.1 K/decade warming trend, as seen from 1998-2015 or from 1999-2014.

14. Line 274: "unexpected behavior" is an odd term-the authors have just outlined that either the slowdown will continue or it won't, so we can expect one or the other, and so there is really nothing unexpected about what is happening, we just don't know which of the two alternatives is happening.

OK, we removed "unexpected".

REVIEWERS' COMMENTS:

Reviewer #2 (Remarks to the Author):

The authors have done an admirable job in revision and I now recommend publication.

Reviewer #3 (Remarks to the Author):

The authors have adequately responded to my comments, and the manuscript is now ready for publication.